# “Key Factor” for Baijiu Quality: Research Progress on Acid Substances in Baijiu

**DOI:** 10.3390/foods11192959

**Published:** 2022-09-21

**Authors:** Yashuai Wu, Yaxin Hou, Hao Chen, Junshan Wang, Chunsheng Zhang, Zhigang Zhao, Ran Ao, He Huang, Jiaxin Hong, Dongrui Zhao, Baoguo Sun

**Affiliations:** 1Key Laboratory of Brewing Molecular Engineering of China Light Industry, Beijing Technology and Business University, Haidian District, No. 11, Fucheng Road, Beijing 100048, China; 2Beijing Laboratory of Food Quality and Safety, Beijing Technology and Business University, Beijing 100048, China; 3Chengde Qianlongzui Distillery Company, Chengde 067400, China; 4Department of Nutrition and Health, China Agriculture University, Beijing 100193, China

**Keywords:** baijiu, acid substances, flavor, function, sensory evaluation

## Abstract

Baijiu is the national liquor of China, which has lasted in China for more than 2000 years. Abundant raw materials, multi-strain co-fermentation, and complex processes make the secrets of baijiu flavor and taste still not fully explored. Acid substances not only have a great influence on the flavor and taste of baijiu, but also have certain functions. Therefore, this paper provides a systematic review for the reported acid substances, especially for their contribution to the flavor and functional quality of baijiu. Based on previous studies, this paper puts forward a conjecture, a suggestion, and a point of view, namely: the conjecture of “whether acid substances can be used as ‘key factor’ for baijiu quality “; the suggestion of “the focus of research on acid substances in baijiu should be transferred to evaluating their contribution to the taste of baijiu”; and the view of “acid substances are ‘regulators’ in the fermentation process of baijiu”. It is worth thinking about whether acid substances can be used as the key factors of baijiu to be studied and confirmed by practice in the future. It is hoped that the systematic review of acid substances in baijiu in this paper can contribute to further in-depth and systematic research on baijiu by researchers in the future.

## 1. Introduction

Baijiu is the national liquor of China, which carries the special emotions of Chinese people; thus, baijiu becomes an indispensable drink for Chinese people in places such as festivals and gatherings. Baijiu is made of sorghum, corn, and other grains [1]. Under the action of Jiuqu (a saccharification-and-fermentation starter) during solid-state fermentation, baijiu is extracted and distilled in a retort (a device for distilling baijiu). After ripening (a kind of storage method that makes baijiu taste better), baijiu is blended by technicians and forms a kind of alcoholic beverage unique to China [2]. At present, baijiu is divided into twelve aroma types, namely strong-aroma (also named as nong-flavor baijiu), jiang-aroma (also named as moutai-aroma or sauce-aroma baijiu), light-aroma, rice-aroma, zhima-aroma (also named as sesame-aroma baijiu), fuyu-aroma, laobaigan-aroma, chi-aroma, herbal-aroma (also named as dong-aroma or drug-aroma baijiu), te-aroma, mixed-aroma (also named as strong-sauce-aroma or miscellaneous-flavor baijiu), and feng-aroma [3,4]. The twelve aroma types of baijiu embody the wisdom of Chinese people. Of note, there is a certain relationship between the twelve kinds of aroma types of baijiu; the specific relationship is shown in Figure 1a. Among them, jiang-aroma, strong-aroma, light-aroma, and rice-aroma types are considered to be the basic aroma types of baijiu. The remaining eight aroma types of baijiu are derived from the four basic aroma types of baijiu. With the innovation of technique, the aroma type is no longer a constraint, and many other aroma types of baijiu have also begun to appear [5]. Baijiu circulating in the market is still dominated by these 12 kinds of aroma types of baijiu.

Baijiu is mainly comprised of ethanol and water (both account for about 98% out of baijiu), and the trace components (about 2% of baijiu) are the material basis for the flavor and functional attributes of baijiu. The above 12 kinds of aroma types of baijiu are mainly distinguished based on different aroma characteristics [6]. As shown, the aroma characteristics depend on the type and concentration of the trace components it contains, specifically the distribution of the flavor substances contained in it. Since 1960, researchers have been working on cracking the flavor and functional codes of baijiu. However, limited by experimental conditions and technical level, it was difficult to analyze and evaluate how many flavor substances and functional substances existed in baijiu. In the past ten years, the emergence, maturity, and application of modern flavor extraction and separation technique, modern flavor detection and analysis technique, molecular sensory omics, and other technical means have greatly promoted the development of research related to trace components and flavor substances in baijiu. Based on the techniques and methods described above, 2020 trace components have been detected in baijiu, and the trace components of different categories are shown in Figure 1b,c. These trace components play an extremely different role in baijiu.

There is no doubt that the flavor and whether it is healthy for the human body after eating or drinking are the core evaluation criteria for the quality of food and beverage, and these two evaluation criteria are also applicable to baijiu. Based on a large number of investigations, it has been found that there is a relatively important link between the flavor and function of baijiu, that is, acid substances. Acid substances can effectively reduce the pungent and bitter taste of baijiu, increase the sweetness of baijiu, and have a very important impact on the taste of baijiu. In addition, acid substances in baijiu can be divided into short-chain acid substances, long-chain acid substances, and other acid substances. The main function of short-chain acid substances is to reduce alcohol intoxication and relieve acute liver damage. The main function of long-chain acid substances is to reduce the risk of cardiovascular disease. The main function of other acid substances is to protect biological systems from the effects of hydroxyl and peroxy free radicals [1,4]. The main functions and structural formulas of important acid substances in baijiu are shown in Figure 2 below. Meanwhile, most acid substances are precursors for the synthesis of esters, and esters are currently considered to be the most important type of trace components that determine the aroma of baijiu. Although there are not many studies that deem acid substances as the linkage trace components for the flavor and function of baijiu, acid substances are undoubtedly a very important type of trace components in baijiu [7]. Based on the above, combined with the research results obtained by different pretreatment methods and detection methods of acid substances in baijiu, this paper systematically sorted out the distribution of 121 acid substances in different aroma types of baijiu. Then, combined with the frequency, concentration, aroma contribution, and other indicators of acid substances in different aroma types of baijiu, 14 kinds of acid substances were found to make an important contribution to the quality of baijiu. In addition, this paper also combed the functions and effects on baijiu fermentation process of important acid substances in baijiu. It is hoped that the systematic review on acid substances in baijiu from this paper can contribute to further in-depth and systematic research on the trace components of baijiu in the future.

## 2. Analytical Methods for Acid Substances in Baijiu

In the past ten years, the continuous enrichment of experimental techniques and methods has enabled researchers to pursue more accurate and in-depth research on flavor substances and functional substances in baijiu. At present, researchers have detected 121 kinds of acid substances in baijiu. Possibly due to the great differences in the physical properties (e.g., solubility and volatility) and chemical properties of different acid substances in baijiu, there is yet no fixed pretreatment method or detection method for acid substances. At this stage, the key descriptions of the pretreatment methods for acid substances in baijiu are listed in Table 1 below [8,9].

As can be seen from Table 1, there are many pretreatment methods for the extraction and separation of acid substances in baijiu, while different methods have their own advantages and disadvantages. As studies showed, the most common pretreatment methods for acid substances in baijiu are direct injection (DI), liquid-liquid extraction (LLE), and liquid-liquid microextraction (LLME) [10]. Compared with other pretreatment methods, the most prominent features for DI and LLME are simple and easy to operate. They were mostly suitable for the detection of acid substances with high concentration in baijiu. The most prominent feature for LLE is the high enrichment rate, which is more helpful for the analysis and detection of low-concentration acid substances in baijiu. Solid phase extraction (SPE), solid phase microextraction (SPME), and stir bar sorptive extraction (SBSE) methods are characterized by a high level of automation and environmental protection. Of note, SPME is applied to obtain volatile acid substances. SBSE is widely conducted to absorb volatile and non-volatile acids substances in baijiu. Therefore, in terms of the difference in physical and chemical properties of acids, it is necessary to comprehensively use various pretreatment methods to better separate and extract the acid substances in baijiu and provide a basis for subsequent analysis [11].

Afterwards, appropriate analytical techniques are selected for the analysis of acid substances in baijiu. After being separated and extracted, based on the different properties of acid substances, a more comprehensive analysis of acid substances in baijiu is conducted by combining different analysis methods. At present, the analytical instruments or techniques applied for analysis on acid substances in baijiu mainly include: mass spectrometry (MS), ion chromatography (IC), ultra-high performance liquid chromatography (UPLC), gas chromatographic-mass spectrometer (GC-MS), and gas chromatography-olfactometry (GC-O). Recently, besides the previous common analytical techniques, new analytical techniques (i.e., gas chromatography-quadrupole time of flight mass spectra (GC-QTOF-MS), liquid chromatography-quadrupole time of flight (LC-QTOF), and two-dimensional gas chromatography-time-of-flight mass spectrometry (GC×GC-TOF-MS)) have begun to be applied and promoted the detection of acid substances in baijiu [12,13,14].

## 3. Verification of Acid Substances in Baijiu

Roughly, the analysis of acid substances in baijiu is mainly divided into three levels. The identification on acid substances in baijiu is the first step, which is also the basis for further clarifying the concentration distribution of acid substances in baijiu and their contribution to baijiu flavor. Based on the above-mentioned pretreatment methods and detection techniques, the current qualitative methods for acid substances in baijiu broadly include: standard comparison method (SCM), spectral library identification (SLI), retention index (RI), and aroma comparative analysis (ACA) [15].

With the continuous enrichment of pretreatment methods, detection techniques, and qualitative analysis methods, the identification for acid substances in baijiu has been greatly promoted. At present, 121 kinds of acid substances have been confirmed in baijiu, mainly based on SLI and SCM. In detail, the qualitative research advance on acid substances in baijiu are shown in Table 2 and Table 3 [16].

As shown in Table 2 and Table 3, it can be seen that acetic acid, propionic acid, butanoic acid, hexanoic acid, pentanoic acid, heptanoic acid, benzenepropionic acid, 2-methylpropionic acid, 4-methylpentanoic acid, and 2-methylbutanoic acid widely exist in various aroma types of baijiu. These acid substances form the basis for the flavor of baijiu. In contrast, most other acid substances are only present in part aroma types of baijiu. For instance, benzoylformic acid, chlorogenic acid, *β*-nitropropionic acid, 3-methoxybutyric acid, 3,4-dihydroxycinnamic acid, gallic acid, and 2,3-dimethyl-2-pentenoic acid mainly exist in strong-aroma baijiu; geranic acid, 9-decenoic acid, 2-hydroxy-2-methyl-propanedioic acid, 17-octadecynoic acid, and 2-ethyl-2-hydroxybutyric acid mainly are verified in jiang-aroma baijiu; and 5-hexenoic acid, undecenoic acid, *cis*-13-docosenoic acid, ethoxyacetic acid, 3-decenoic acid, 4-heptenoic acid, and 2-hydroxydodecanoic acid are mainly present in light-aroma baijiu. The unique acid substances in other aroma types of baijiu are slightly single: methyltartronic acid is found in fuyu-aroma; vaccenic acid and oxalic acid dihydrate are verified in zhima-aroma baijiu [17].

The distribution of acid substances in different baijiu aroma types is shown in Figure 3. Combined with Figure 3 and the above mentioned, it can be found that the three aroma types of baijiu (i.e., strong-aroma, jiang-aroma, and light-aroma) are very rich in acid substances. Furthermore, they are currently the three most representative aroma types of baijiu. Therefore, it can be speculated that acid substances have a huge impact on baijiu quality.

## 4. Aroma Evaluation for the Acid Substances in Baijiu

The preceding section sorted out the distribution of 121 acid substances in different aroma types of baijiu. It can be seen that there is variance in the distribution of these 121 acid substances in different aroma types of baijiu. Then, through evaluating the frequency, concentration, aroma contribution, and other indicators of acid substances in different aroma types of baijiu, 14 kinds of acid substances were verified that made an important contribution to the quality of baijiu. The literature survey found that researchers combined theory and production to further conduct more in-depth research on these 14 acid substances that have an important impact on the quality of baijiu. Specifically, the concentration of these 14 acid substances in baijiu was measured with external standard method (ESTD), internal standard method (ISTD), stable isotope dilution analysis (SIDA), etc. Furthermore, the aroma expression of these 14 acid substances was evaluated by odor activity values (OAVs), aroma extract dilution analysis (AEDA), odor specific magnitude estimation (Osme), and other methods [20,23]. Afterwards, sensory evaluation was introduced to verify the important acid substances that affect the aroma profile of baijiu.

The evaluation for the aroma expression of important acid substances in different baijiu is shown in Table 4, Table 5 and Table 6. As Table 4, Table 5 and Table 6 exhibits, it is observed that the aroma thresholds of acetic acid, propionic acid, heptanoic acid, and decanoic acid are greater than 10,000 μg/L. Therefore, this also means that only when the concentration of acid substances in baijiu is high can it contribute to the flavor of baijiu [44].

Studies have shown that the concentration of acid substances and esters in baijiu is higher, but the aroma threshold of acid substances is significantly higher than that of esters. Thus, the OAV values of acid substances in Table 4–6 are extremely lower than those of esters in baijiu (e.g., ethyl hexanoate OAV = 6867, ethyl octanoate OAV = 3447, ethyl butanoate OAV = 1310, and ethyl 3-methylbutanoate OAV = 1144). Even the OAV value of acid substances in baijiu is not as good as that for part of sulfur-containing compounds (e.g., dimethyl trisulfide OAV = 2990) with very low concentration. Whilst it is also worth noting that although hexanoic acid has an OAV value of 3~35 in various baijiu samples, hexanoic acid combining with ethanol will generate ethyl hexanoate (OAV = 6867), thereby, exhibiting crucial effect on the flavor adjustment of the baijiu. In particular, from the perspective of metabolism, the concentration of acid substances can determine the concentration of esters, and esters are typically considered to be the most important type of flavor substances in baijiu. Therefore, the balance of acids and esters will become a very important key control point for the quality of baijiu. A large number of investigations on the flavor of baijiu have found that there is a certain balance between acid substances and esters in various types of baijiu. The reason is that acid substances are important precursors of esters.

Of note, acid substances are produced during the metabolism of ethanol. In general, *Saccharomyces* generates ethanol under the action of pyruvate decarboxylase and alcohol dehydrogenase through the glycolysis pathway under anaerobic conditions. During the production of ethanol, some other metabolic by-products are also produced, such as 2-methyl-1-propanol, 2-methyl-1-butanol, 3-methyl-1-butanol, etc. These by-products are the precursors of acid substances in baijiu. Alcohol dehydrogenases and aldehyde dehydrogenases catalyze 2-methyl-1-propanol, 2-methyl-1-butanol, and 3-methyl-1-butanol, and other substances to the corresponding acids. The most regularly detected acid substances in baijiu (acetic acid, propionic acid, butanoic acid, hexanoic acid, pentanoic acid, heptanoic acid, nonanoic acid, octanoic acid, decanoic acid, 2-methylpropionic acid, and 3-methylbutanoic acid) are produced by the above-mentioned ethanol metabolism pathway. In detail, acetic acid is produced in a metabolic pathway by *Acetobacter aceti*. Lactic acid can be produced during the fermentation of *lactic acid bacteria*. Furthermore, acetic acid and lactic acid can be used as substrates for butanoic acid and hexanoic acid. The main pathway for the formation of butyrate is the reverse *β*-oxidation (microbes use ethanol and acetic acid as substrates to form butyryl-CoA, which is then generated by CoA-transferase or phosphate butyryl-transferase and butyric acid kinase). On the basis, the production of caproic acid and caprylic acid is comparable to the above pathway. All in all, acid substances are primarily produced by various microorganisms using a variety of acetyl-transferase systems. Afterwards, ethyl esters are produced by acid substances and ethanol under the action of esterase (e.g., carboxylic acid hydrolase family, lipase, esterase, and cutinase).

As studies showed, the esters formed by short-chain acid substances and ethanol are the main trace components of baijiu, and principally express fruity or floral aroma, such as ethyl acetate, ethyl hexanoate, ethyl butyrate, etc. Moreover, these basic acid substances will undergo chemical changes to generate 2-methylpropanoic acid, 3-methylbutanoic acid, tetradecanoic acid, sorbic acid, lactic acid, benzoic acid, 3-phenylpropanoic acid, etc., which will influence the flavor of baijiu. The important acid substances and their derived esters in baijiu are shown in Figure 4.

Therefore, it can be found that acid substances are not only the indispensable flavor substances in baijiu, but also the precursor substances for the synthesis of various ethyl ester compounds, and then become the “behind force” that affects the aroma of baijiu.

Of note, more study is aimed to dig out the support of acid substances to baijiu aroma. While the contribution of acid substances to the taste of baijiu is neglected, in contrast, the research on acid substances in wine and fruit wine is relatively systematic [51,52,53]. Obviously, as an alcoholic beverage, the process of “smell the fragrance” before drinking is the distinctive feature of baijiu, and the flavor substance is also the material basis for the aroma of baijiu. However, it is undeniable that in addition to enjoying the aroma, the taste of baijiu is also particularly important. During the process of drinking baijiu, if the taste does not meet consumer expectations, it will also affect the acceptance of the product.

In addition, in the field of sensory evaluation, there are also some studies on the influence of acid substances on the sensory of baijiu. Interestingly, combining the aroma descriptions of acid substances (e.g., acid, fruit, pungent, vinegar, fat, pungent, silage, soy, butter, cheese, sour, cheese, oil, and pungent), it can be found that these descriptions are mostly related to the taste. However, because sensory words are very local, it is difficult to describe them with a generic description. The above problems have also blurred the influence of acid substances on the taste of baijiu. Most of them are non-professional research conducted by baijiu factories, rather than research conducted by researchers using modern science and technology. No matter, through those sensory words with local characteristics, it can still find the important influence of acid substances on the quality of baijiu. The evaluation requirements for the taste of high-quality baijiu are: sweet and smooth in the mouth, soft and mellow, not dry or spicy, natural irritation, and no pain in the mouth; when drinking, there is a certain thick and round feeling in the mouth, no water taste, the aroma and baijiu taste should be coordinated, and the aroma should not be greater than the taste of baijiu, nor should the taste of baijiu be greater than the aroma; it has obvious sweetness, no bitterness, clean aftertaste, no astringency and saltiness, and no obvious nausea after drinking. Although this evaluation standard is too abstract, it can still be seen that acid substances play an irreplaceable role in baijiu. In fact, choosing a good baijiu is like choosing an orange with a suitable sugar-to-acid ratio, or *Coca-Cola*. There is no doubt that acids play a crucial role, and too much or too little concentration is not perfect [54].

It is a pity that researchers still deem acid substances as flavor substances; they have carried out more research on their functions. In the future, the focus of research on acid substances should be transferred to evaluating their contribution to the taste of baijiu.

## 5. Function of Important Acid Substances in Baijiu

Baijiu is a commodity, and comfort after drinking is very important. The comfort of baijiu after drinking is also the most important manifestation of the functionality of baijiu. Generally speaking, the drinking comfort of baijiu has two meanings: one is the comfort during the drinking process, that is, the comprehensive feeling of the color, aroma, and taste of the baijiu during drinking; the second is the comfort level after drinking, the physiological response after drinking. High quality baijiu has the characteristics of no dizziness and no thirst after drinking. Internationally, long-term detection of brain nerve activity, state, function, blood flow, and metabolism after drinking alcohol is used to evaluate the variances in the comfort level of different alcohol after drinking, so as to determine the characteristics of high-quality baijiu [55,56,57,58,59,60].

Many years ago, “blending of flavor” baijiu produced by some manufacturers appeared in the Chinese market. After drinking, consumers felt thirst, dizziness, and took other side effects, and the body recovered slowly, thereby affecting the subsequent work and rest [61]. It has been found out that the main reason was that the proportion of acid, ester, alcohol, and aldehyde in the baijiu was not coordinated, especially the lack of acid and ester [56]. Therefore, baijiu contains coordinated acids and esters, which is not only a requirement for flavor quality, but also has a certain function. Acid substances are the important material basis of baijiu quality. Of note, acid substances (including esters formed by acid substances and ethanol) can not only affect the color, aroma, and taste of baijiu, but also make baijiu have a positive impact after drinking [62,63,64]. The structure diagram of functional acid substances in baijiu and their main functional descriptions are shown in Figure 2 [65,66,67].

### 5.1. Short-Chain Acid Substances

The short-chain acid substances in baijiu are mainly acetic acid, butyric acid, lactic acid, caproic acid, etc. These four kinds of short-chain acid substances can effectively reduce the alcohol damage caused by drinking. Acetic acid and ethyl acetate can significantly reduce alcohol intoxication, and reduce intoxication by affecting ethanol metabolism, and have a positive effect on acute alcoholic liver injury [68,69]. Acetate, propionate, and butyrate can reduce intestinal inflammation through two signaling pathways (i.e.,GPCRs activation pathway and HDACs inhibition pathway), and maintain the normal physiological state of the colon [70]. Butyric acid is also a cancer therapeutic agent, which can antagonize HDACs, keep chromatin in an open state with high transcriptional activity, increase the expression of P_21_ gene, stop the cell cycle in G0 phase, and inhibit colon cancer cell proliferation.

Baijiu also contains some active branched short-chain acid substances. L-lactic acid is an organic acid produced by microbial fermentation with corn as raw material. It can stabilize the balance of micro-ecology in the body, maintain an acidic environment in the stomach, and inhibit the growth of pathogenic bacteria while promoting the growth of *bifidobacterial* [71,72]. L-malic acid has the effect of relieving cough and asthma. Gallic acid has antibacterial and antiviral effects and has inhibitory effects on pathogenic bacteria such as *Staphylococcus aureus, Sarcines, Neisseria, Pseudomonas aeruginosa, Shigella flexneri*, and other pathogens in vitro [73]. Gallic acid can also up-regulate p53 protein, induce apoptosis of cells containing HPV gene sequence, inhibit the reproduction of HPVep, and have good selectivity for HPV-positive cells, which provides the possibility for the research of new antiviral drugs [74]. Butanedioic acid has antibacterial and antiulcer effects (inhibits gastric juice secretion and dilates gastric muscle for antiulcer effect) [75,76,77].

### 5.2. Long-Chain Acid Substances

The corn in the brewing raw material contains more phytic acid, which can be fermented to produce cyclohexanol, thereby, alleviating liver cirrhosis, hepatitis, fatty liver, high blood cholesterol, and other diseases [78]. Linoleic acid is an unsaturated acid that binds cholesterol and promotes its excretion, while also lowering blood levels of low-density lipoprotein (LDL), very low-density lipoprotein (VLDL), and triglycerides, and increasing high-density lipoprotein (HDL) levels. In addition, HDL has the effects of modifying microcirculation, softening blood vessels, and reducing the risk of cardiovascular disease [79,80].

### 5.3. Other Acid Substances

Ferulic acid can protect biological systems from hydroxyl and peroxy free radicals. In addition, it was also found that ferulic acid has anti-platelet aggregation, inhibits platelet 5-hydroxytryptamine (5-HT) release, inhibits platelet thromboxane (TXA2) production, enhances prostaglandin activity, analgesia, and relieves vasospasm. Moreover, some scholars have confirmed that ferulic acid can exert anti-angiogenesis and anti-tumor (melanoma) effects by inhibiting fibro-blast growth factor receptor 1 (FGFR1) [81].

Although attention has been paid to the function for acid substances in baijiu, more studies were based on function evaluation for acids; the research on their activities in baijiu matrix and their contribution to the overall function of baijiu was uncovered.

## 6. The Role of Acid Substances during Baijiu Fermentation Process

The above is the importance of the influence of acid substances on the flavor and functional quality of baijiu. It is also worth exploring how the acid substances in baijiu are produced and whether it will affect the fermentation process of baijiu.

Baijiu fermenting technique is a long-established food processing technique, and “Jiuqu is a saccharification starter” is a characteristic feature of the production process. As a “multi-bacteria and multi-enzyme” microbial product, Jiuqu is an important carrier of baijiu brewing microorganisms, functional enzymes, flavors, and their precursors. The variety and abundance of its three lines of “bacteria, enzymes and flavor” determine the unique flavor quality and style characteristics of baijiu. One of the most important is bacteria, which is the source of flavor substances [82,83].

The general Jiuqu applied in the baijiu industry mainly includes Daqu and Xiaoqu. The specific production process is shown in Figure 5 below. Daqu, also known as Kuaiqu or Zhuanqu, as shown in Figure 5, uses wheat, barley, or peas as raw materials, is crushed, kneaded with water, pressed into a brick-like qu billet, and enriched with various microorganisms from nature. According to its culture temperature, it can be divided into high temperature Daqu, medium temperature Daqu, and low temperature Daqu. Xiaoqu mostly uses rice flour or rice bran as raw materials, and sometimes adds Chinese herbal medicine as auxiliary materials, and is cultivated under the conditions of temperature and humidity control. It is called “Xiaoqu” because of its small size [84,85,86,87].

After the Jiuqu is mixed with the raw materials, the microorganisms within Jiuqu begin to “work”. As the main raw material for baijiu fermentation, the starch contained in sorghum cannot be directly utilized by most bacteria and yeasts and needs to be hydrolyzed into fermentable sugars by *α*-amylase and saccharification enzymes produced by molds.

Therefore, in the early stage of fermentation, fungi play an important role in the fermenting process. In particular, molds are regarded as the main source of saccharification and degradation power of Jiuqu, providing saccharification power, liquefaction power, and protein decomposition ability for the fermentation of baijiu. For example, the relative abundance of *Aspergillo, Mucor, Rhizopus, Mucor racemosus*, and *Thermoascus crustaceus* in the early stage of baijiu fermentation is relatively high. Thus, molds have an important influence on the aroma and taste of baijiu [85,88].

After that, various types of bacteria begin to use glucose for fermentation, and the relatively concentrated bacteria are *Bacillus*, *Weissella*, *Lactobacillus*, etc. [89]. Of note, the fermentation acidity determined by acid substances is closely related to the change on the number of microorganisms, especially for bacteria. Specifically, fermentation acidity has a positive correlation with the abundance of *Bacillus*. While along with the increase of fermentation acidity, the abundance of *Bacillus* decreased from the initial 2.09% to 0.18%, and then the fermentation acidity also gradually decreased [90,91]. Thus, along with the decrease on the acidity of the fermentation environment, the abundance of yeast begins to increase. Yeast has 13 enzyme systems (e.g., coenzyme nicotinamide of oxidoreductase, coenzyme of decarboxylase, and reduced glutathione and ribonuclease), which can effectively catalyze the esterification reaction and effectively reduce the acidity of the fermentation environment at a later stage [92,93]. During the change process of fermentation acidity, *Lactobacillus* can promote Maillard reaction and maintain the micro-ecological environment of fermentation. *Weissella* decomposes glucose to produce CO_2_ and lactic acid by heterofermentation [94]. Furthermore, lactic acid is beneficial to the formation of ethyl lactate and the stability of baijiu quality [95]. Thus, acid substances were not only an important metabolite of microorganism, but also played an important role in regulating the metabolism of microorganism, which determined the quality of baijiu [85].

To sum up, it can be found that strict control of acidity during the fermentation process helps to regulate and balance the composition and quantity of microorganisms, subsequently, ensuring the quality of baijiu [96]. Meanwhile, the acid substances produced will also pass through the esterification reaction to produce esters and devote to the aroma of baijiu. Therefore, it is speculated that pH during fermentation is the core of fermentation control. In the future, the realization of precise control of multi-strain and pure microbial enhanced fermentation is a direction for Chinese baijiu to transform from traditional fermentation to modern fermentation [97]. In the future, we can customize the flavor of baijiu by customizing microorganisms, and ultimately realize controllable microorganisms, controllable metabolism, and controllable flavor, which is also the future prospect of the baijiu industry. It can be seen that acid substances will play the crucial role of “regulator” in the fermentation process of baijiu.

## 7. Conclusions and Outlook

Baijiu has a history of more than 2000 years in China. Up to now, with the rapid development of analysis and detection techniques, a large number of trace components have been detected in baijiu, including 121 kinds of acid substances. From the above review, clear points can be drawn, i.e., acid substances in baijiu not only make important contributions to the aroma, taste, and function of baijiu, but also play an important role in regulating the metabolism of other trace components in baijiu, thereby affecting the quality of baijiu. Therefore, the study of acids in baijiu as the core of the interaction between the flavor and taste of baijiu may be one of the important directions for future research. Of note, exploring the changes in microorganisms during baijiu fermentation is extremely complex. In order to better reveal the relationship between microbial metabolites and flavor substances in the process of baijiu fermentation, acid-producing microorganisms can be used as the starting point to explore the law, which is of great significance to the production practice of baijiu. Finally, it is worth thinking about whether acid substances can be used as the key factors of baijiu to be studied and confirmed by practice in the future.

## Figures and Tables

**Figure 1 foods-11-02959-f001:**
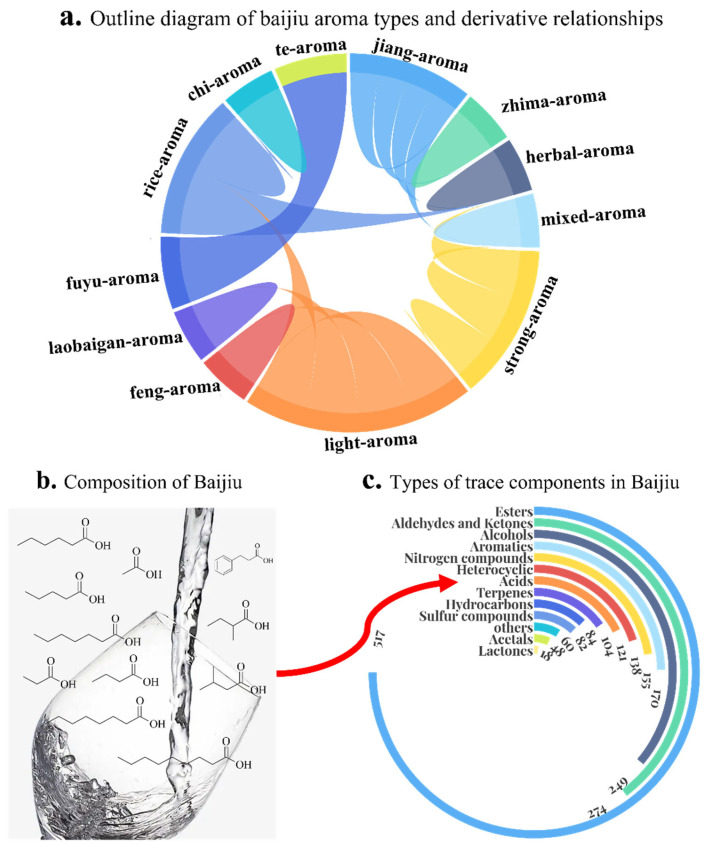
General situation of research on aroma types and trace components in baijiu. Among them, figure (**a**) is about outline diagram of baijiu aroma types and derivative relationships; figure (**b**) is about the trace components of baijiu, and the structural formula in the figure is represented by acid substances; figure (**c**) is about types of trace components in Baijiu.

**Figure 2 foods-11-02959-f002:**
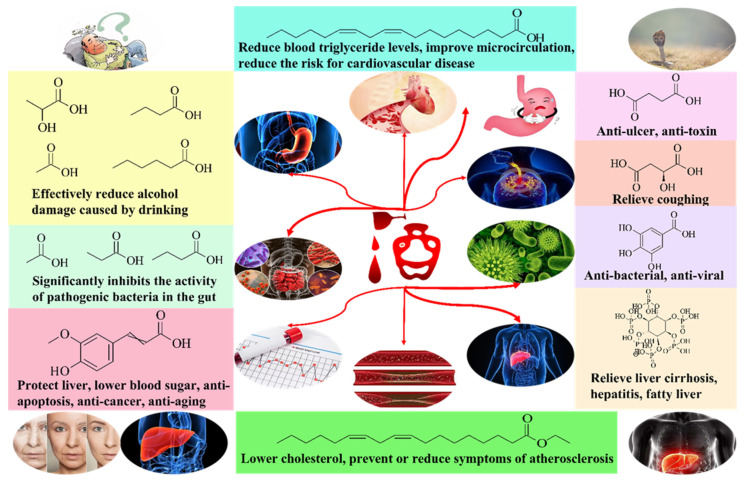
Main functions and structural formulas of important acid substances in baijiu.

**Figure 3 foods-11-02959-f003:**
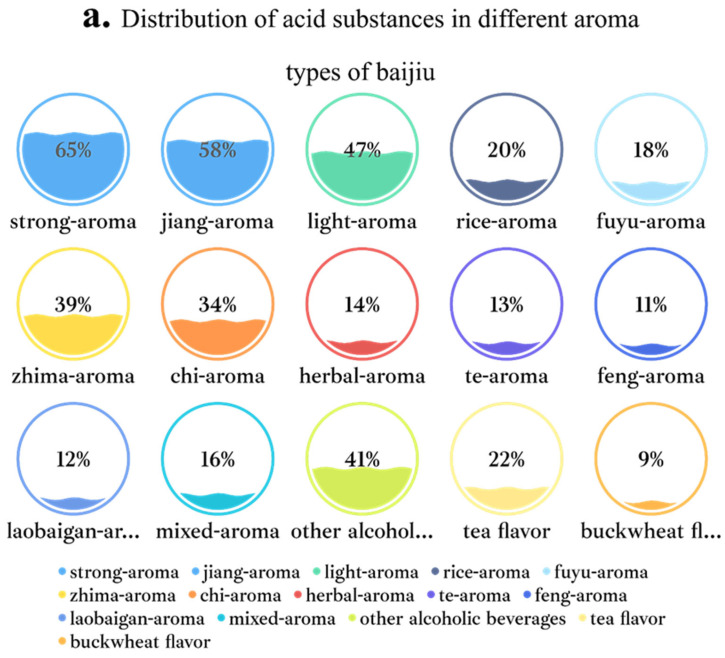
Intuitive diagram of identification on acid substances in different aroma types of baijiu. Among them, figure (**a**) is about distribution of acid substances in different aroma types of baijiu. The ratio of the numbers in the figure represents: the proportion of acid substances in this fragrance to 121 acid substances; figure (**b**) is about relationship between 121 kinds of acid substances and different aroma types of baijiu.

**Figure 4 foods-11-02959-f004:**
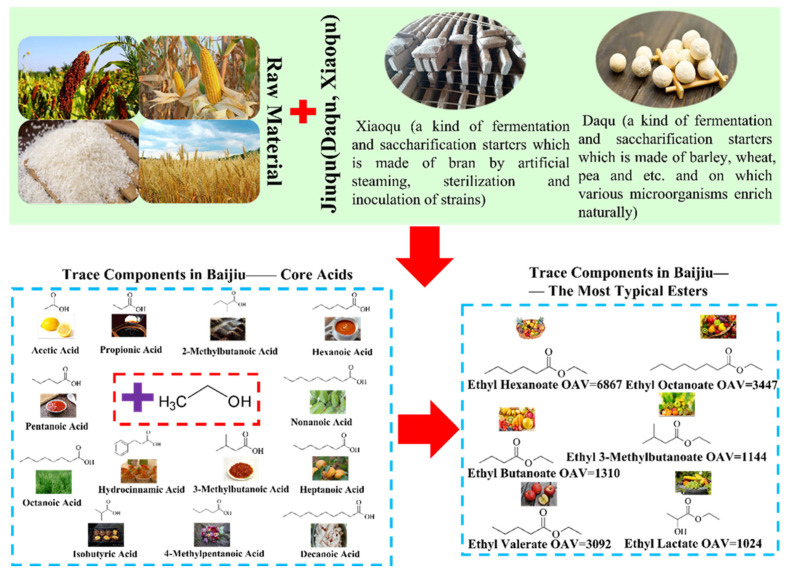
Important acid substances and their derived esters in baijiu.

**Figure 5 foods-11-02959-f005:**
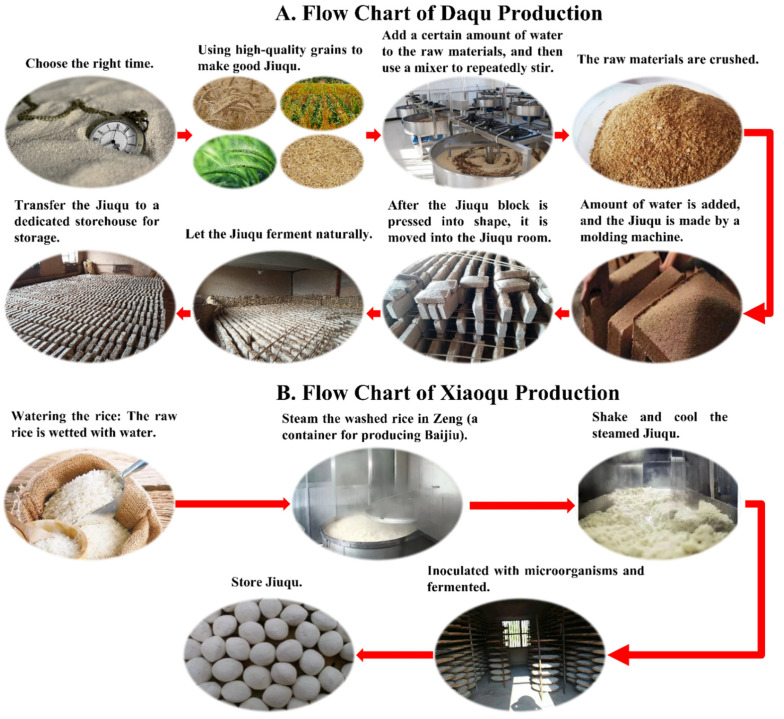
Flow chart of Jiuqu production. Figure (**A**) is the flow chart of Daqu production. Figure (**B**) is the flow chart of Xiaoqu production.

**Table 1 foods-11-02959-t001:** The key description of the pretreatment method of acid substances in baijiu.

Pretreatment Methods	Key Description and Advantages	Disadvantages
Direct injection (DI)	This pretreatment method has the characteristics of simple operation and little loss for trace components.	This method is difficult to detect trace compounds.
Liquid-liquid extraction (LLE)	This pretreatment method has the peculiarities of wide application range and high enrichment rate for trace components.	This method requires a large number of samples and extractants and takes a long time; it is not suitable for detecting compounds with low boiling points.
Liquid-liquid microextraction (LLME)	This pretreatment method has the features of less amount of extractant and short pretreatment time.	Many compounds cannot be detected by this method.
Solid phase extraction (SPE)	This pretreatment method can simultaneously complete the enrichment and purification of trace components in baijiu, and has the characteristics of high sensitivity and good reproducibility.	This method is not effective for the detection of highly polar compounds such as fatty acids.
Solid phase microextraction (SPME)	This pretreatment method has the specialties of simple operation, green environmental protection, and short time-consuming.	This method is not effective for the detection of highly polar compounds such as fatty acids.
Stir bar sorptive extraction (SBSE)	The method has the features of large adsorption capacity, high sensitivity, and good recovery rate, and can simultaneously complete the extraction and enrichment of trace components while stirring by itself. This pretreatment method is suitable for enriching trace components in aqueous samples.	There are fewer types of adsorption coatings, higher costs, and certain disproportionation effects.
Simultaneous distillation and extraction (SDE)	The method has the characteristics of less solvent consumption and high enrichment rate, and can be simultaneously distilled and extracted for trace components. This pretreatment method is mostly used for the enrichment of trace volatile components in complex matrix.	The heating temperature of this method is high, which affects the structural analysis of the flavor components for baijiu.
Supercritical fluid extraction (SFE)	The method has the specialties of environmental friendliness, high automation, and simple extraction process. It can well enrich the substances with high boiling point, low volatility, and easy pyrolysis.	It is difficult to separate and recover the entrainer from the extract, and the entrainer may remain in the extract.

**Table 2 foods-11-02959-t002:** Identification overview for acid substances in different types of baijiu.

Name	CAS Number	Pre-Processing Method	Identification Method	Strong-Aroma	Jiang-Aroma	Light-Aroma	Rice-Aroma	Fuyu-Aroma	Zhima-Aroma	Chi-Aroma	Herbal-Aroma
5,8,11,14-tetracosic acid	1191-85-1	SDE	MS, SLI	√	—	—	—	—	—	—	—
2,2-dimethyl-hexanoic acid	813-72-9	SDE	MS, SLI	√	—	—	—	—	—	—	—
(E)-2-methyl-2-pentenoic acid	16957-70-3	LLE	MS, SLI	√	—	—	—	—	—	—	—
ethyl 2-phenylhexanoate	2901-28-2	SPME	MS, SLI	√	—	—	—	—	—	—	—
ethyl isopentyl succinate	28024-16-0	LLE	MS, RI, SLI	—	√	√	—	—	—	—	—
formic acid	64-18-6	SPME, LLE	MS, IC, SLI	√	√	√	√	—	√	√	—
acetic acid	64-19-7	LLE, SDE, SPME, SAFE, DI	MS, RI, ACA, SCM, IC, SLI	√	√	√	√	√	√	√	√
propanoic acid	1979/9/4	LLE, SDE, DI, SPME	MS, SCM, RI, ACA, IC, SLI	√	√	√	√	√	√	√	√
2-methylpropanoic acid	79-31-2	LLE, DI, SDE, SPME	MS, SCM, RI, ACA, SLI	√	√	√	√	√	√	√	√
butanoic acid	107-92-6	LLE, SPME, DI	MS, SCM, RI, ACA, IC, SLI	√	√	√	√	√	√	√	√
2-methylbutanoic acid	116-53-0	LLE, SPME	MS, SCM, RI, ACA, SLI	√	√	√	√	√	√	—	√
3-methylbutanoic acid	503-74-2	LLE, SPME	MS, SCM, RI, ACA, SLI	√	√	√	√	√	√	—	√
tannic acid	1401-55-4	LLE, SDE, SPME, DI	MS, SCM, RI, SLI	√	√	√	√	—	√	√	—
2-ethylbutyric acid	1988/9/5	DI, SPME, LLE	MS, SCM, MS	√	√	√	—	—	—	√	—
pentanoic acid	109-52-4	LLE, SPME, DI, SDE	MS, SCM, RI, ACA, SLI	√	√	√	√	√	√	√	√
2-methylpentanoic acid	97-61-0	LLE	MS, SCM, RI, SLI	√	√	√	√	√	√	—	—
3-methylpentanoic acid	105-43-1	LLE	MS	√	—	√	—	—	—	—	—
4-methylpentanoic acid	646-07-1	LLE, DI, SPME	MS, RI, ACA, SLI	√	√	√	√	√	√	—	—
hexanoic acid	142-62-1	LLE, SPME, DI, SPE, SDE	MS, SCM, RI, ACA, SLI	√	√	√	√	√	√	√	√
2-methylhexanoic acid	4536-23-6	LLE	MS, SLI	√	—	—	—	—	—	—	—
5-methylhexanoic acid	628-46-6	LLE	MS, SLI	√	√	—	—	—	√	—	—
heptanoic acid	111-14-8	LLE, SPME, DI, SPE	MS, SCM, RI, ACA, SLI	√	√	√	√	√	√	√	√
octanoic acid	124-07-2	LLE, SPME, DI, SDE	MS, SCM, RI, ACA, SLI	√	√	√	√	√	√	√	√
nonanoic acid	112-05-0	LLE, SPME	MS, SCM, RI, ACA, SLI	√	√	√	√	√	√	√	√
decanoic acid	334-48-5	LLE, SDE, DI, SPME	MS, SCM, RI, ACA, SLI	√	√	√	√	√	√	√	√
undecanoic acid	112-37-8	LLE	MS, SCM, RI, SLI	√	—	√	—	—	—	—	—
lauric acid	143-07-7	LLE, SDE, SPME	MS, SCM, RI, SLI	√	√	√	—	—	√	—	—
tridecanoic acid	638-53-9	LLE	MS, SCM, RI, SLI	√	√	√	—	—	√	—	—
tetradecanoic acid	544-63-8	LLE, SPE, SPME	MS, SCM, RI, SLI	√	√	√	—	—	√	√	—
pentadecanoic acid isomer	1002-84-2	LLE	MS, SLI	√	√	√	—	—	—	—	—
pentadecanoic acid	1002-84-2	LLE	MS, SCM, RI, SLI	√	√	√	—	—	√	—	—
hexadecanoic acid	1957/10/3	LLE, SPME, SDE, DI	MS, SCM, RI, SLI	√	√	√	—	—	√	√	—
heptandecanoic acid	506-12-7	LLE, SDE	MS, SLI	√	√	—	—	—	√	—	—
octadecanoic acid	107-93-7	LLE, SDE, DI, SAFE	MS, SCM, RI, SLI	√	√	—	—	—	√	√	—
methacrylic acid	79-41-4	LLE	MS, SLI	√	—	—	—	—	—	—	—
trans-crotonic acid	107-93-7	LLE	MS, SCM, RI, SLI	√	—	√	—	—	√	—	—
3-methyl-2-butenoic acid	541-47-9	LLE	MS, SCM, RI, ACA, SLI	√	√	√	—	—	—	—	—
2-methyl-2-pentenoic acid	3142-72-1	SPME, LLE	MS, RI, SLI	√	—	—	—	—	—	—	—
2,3-dimethyl-2-pentenoic acid	122630-51-7	LLE	MS, SLI	√	—	—	—	—	—	—	—
(E)-2-hexenoic acid	13419-69-7	LLE	MS, SCM, RI, SLI	√	√	√	—	—	√	—	—
(E)-3-hexenoic acid	1577-18-0	LLE	MS, SCM, RI, SLI	√	—	√	—	—	—	—	—
5-hexenoic acid	1577-22-6	SPME, SDE, LLE	MS, SLI	—	—	√	—	—	—	—	—
sorbic acid	110-44-1	SPME	MS, SLI	—	√	√	—	—	√	√	—
2-heptenoic acid	10352-88-2	LLE	MS, SLI	√	—	√	—	—	—	—	—
2-octenoic acid	1871-67-6	LLE	MS, SLI	√	—	√	—	—	—	—	—
geranic acid	459-80-3	LLE	MS, SLI	—	√	—	—	—	—	—	—
9-decenoic acid	14436-32-9	LLE	MS, RI, SLI	—	√	—	—	—	√	—	—
undecenoic acid	112-38-9	SPME, SDE, LLE	MS, SLI	—	—	√	—	—	—	—	—
9-hexdecenoic acid	2091-29-4	LLE	MS, SLI	—	√	—	—	—	√	—	—
palmitoleic acid	373-49-9	DI, LLE	MS, SCM, SLI	√	√	—	—	—	√	√	—
(E)-9-hexadecenoic acid	10030-73-6	LLE	MS, SLI	—	√	—	—	—	—	—	—
linolenic acid	463-40-1	LLE	MS, SLI	√	—	—	—	—	—	—	—
(E)-9,12,13-trihydroxy-10-octadecaenoic acid	135214-49-2	LLE	MS, SCM, SLI	√	√	—	—	—	—	—	—
linoleic acid	60-33-3	LLE, DI	MS, SCM, SLI	√	√	√	—	—	√	√	—
oleic acid	112-80-1	LLE, SDE	MS, RI, SCM, SLI	√	√	—	—	—	√	√	—
vaccenic acid	506-17-2	LLE	MS, SLI	—	—	—	—	—	√	—	—
(Z)-13-docosenoic acid	112-86-7	LLE	MS, SLI	—	—	√	—	—	—	—	—
oxalic acid	144-62-7	LLE, SDE	IC	—	√	√	—	—	—	√	—
oxalic acid dihydrate	6153-56-6	SPME	MS, SLI	—	—	—	—	—	√	√	—
fumaric acid	110-17-8	SPME	MS, IC, SLI	—	—	—	—	—	—	√	—
pimelic acid	111-16-0	SPME	MS, SLI	—	—	—	—	—	—	√	—
octanedioic acid	505-48-6	LLE, SPME	MS, SLI	—	—	—	—	—	—	√	—
azelaic acid	123-99-9	SPME	MS, SLI	—	—	—	—	—	—	√	—
pathalic acid	88-99-3	SPME	MS, SLI	—	—	—	—	—	—	√	—
1-propene-1,2,3-tricarboxylic acid	499-12-7	SPME	MS, SLI	—	—	—	—	—	—	√	—
citric acid	77-92-9	DI, SPME	MS, SLI	—	—	—	—	—	—	√	—
lactic acid	50-21-5	LLE, DI, SPME	MS, RI, IC, SLI	√	√	√	√	√	√	√	—
(R)-2-hydroxypropionic acid	10326-41-7	SPME	MS, SLI	—	√	—	—	—	—	—	—
2-hydroxyisocaproic acid	10303-64-7	LLE	MS, SLI	—	√	—	—	—	—	—	—
3-hydroxy-dodecanoic acid	1883-13-2	LLE	MS, RI, SLI	√	—	—	—	—	—	—	—
2-hydroxytetradecanoic acid	2507-55-3	SPME	MS, RI, SLI	—	—	—	—	—	—	—	—
benzoic acid	65-85-0	LLE, SPME	MS, SCM, RI, IC, SLI	√	√	√	√	√	√	—	√
vanillic acid	121-34-6	LLE, DI, SPE	MS, SLI	√	—	—	—	—	—	—	—
3-hydroxy-2-methylbenzoic acid	603-80-5	DI	MS, SLI	√	—	—	—	—	—	—	—
syringic acid	530-57-4	LLE, SPE	MS, SLI	—	√	—	—	—	—	—	—
gallic acid	149-91-7	SPE	MS, SLI	√	—	—	—	—	—	—	—
phenylacetic acid	103-82-2	LLE, SAFE	MS, RI, SCM, SLI	√	√	—	√	—	√	—	√
coumaric acid	501-98-4	SAFE	MS, SLI	√	—	—	—	—	—	—	—
3-phenylpropanoic acid	501-52-0	LLE, DI, SPME	MS, SLI, RI	√	√	√	√	√	√	—	—
3,4-dihydroxycinnamic acid	331-39-5	SPE	MS, SLI	√	—	—	—	—	—	—	—
ferulic acid	1135-24-6	LLE, SAFE, SPE	MS, SLI	√	√	√	—	—	—	√	—
benzoyl-formic acid	611-73-4	LLE	MS, SLI	√	—	—	—	—	—	—	—
ethoxy-acetic acid	627-03-2	LLE	MS, SLI	—	—	√	—	—	—	—	—
pyruvic acid	127-17-3	SPME	MS, IC, SLI	√	√	√	—	—	√	√	—
chlorogenic acid	327-97-9	SAFE	MS, SLI	√	—	—	—	—	—	—	—
*β*-nitro-propionic acid	504-88-1	LLE	MS, RI, SLI	√	—	—	—	—	—	—	—
icosanoic acid	506-30-9	LLE	MS, SCM, SLI	—	—	—	—	—	—	√	—
docosanedioic acid	505-56-6	LLE	MS, SCM, SLI	—	—	—	—	—	—	√	—
2-hydroxy-2-phenylpropionic acid	13113-71-8	LLE	MS, SLI	—	√	—	—	—	—	—	—
2-hydroxy-2-methyl-propanedioic acid	595-98-2	LLE	MS, SLI	—	√	—	—	—	—	—	—
3-(4-(benzyloxy)phenyl)-propanoic acid	50463-48-4	LLE	MS, SLI	—	√	—	—	—	—	—	—
(R)-alpha-methoxy-phenylacetic acid	3966-32-3	LLE	MS, SLI	—	√	—	—	—	—	—	—
2-hydroxy-2-methylbutanoic acid	3739-30-8	LLE	MS, SLI	—	√	—	—	—	—	—	—
17-octadecynoic acid	34450-18-5	LLE	MS, SLI	—	√	—	—	—	—	—	—
3-decenoic acid	53678-20-9	LLE	MS, SLI	—	—	√	—	—	—	—	—
4-heptenoic acid	54340-70-4	LLE	MS, SLI	—	—	√	—	—	—	—	—
2-hydroxydodecanoic acid	2984-55-6	LLE	MS, SLI	—	—	√	—	—	—	—	—
methyltartronic acid	80-69-3	SPME	MS, RI, SLI	—	—	—	—	√	—	—	—
*cis*-5-dodecenoic acid	2430-94-6	SPME	MS, SLI	—	√	—	—	—	—	—	—
(Z, Z, Z)-8, 11, 14-eicosatrienoic acid	1783-84-2	SPME	MS, SLI	—	√	—	—	—	—	—	—
2-ethyl-2-hydroxybutyric acid	3639-21-2	SPME	MS, SLI	—	√	—	—	—	—	—	—
3-methoxybutyric acid	10024-70-1	LLE	MS, SLI	√	—	—	—	—	—	—	—
(DL)-serine	302-84-1	LLE	MS, SLI	—	√	—	—	—	—	—	—
glutaric acid	110-94-1	LLE	IC	√	—	—	—	—	—	—	—

“—” means no relevant data from reference; “√” means have relevant data from reference; The data of Table 2 are collectively derived ([1,5,6,7,8,9,16,17,18,19,20,21,22,23,24,25,26,27,28,29,30,31,32,33,34,35,36,37,38,39,40,41,42,43]). Direct injection (DI), Liquid-liquid extraction (LLE), Liquid-liquid microextraction (LLME), Solid phase extraction (SPE), Solid phase microextraction (SPME), Stir bar sorptive extraction (SBSE), Simultaneous distillation and extraction (SDE), Supercritical fluid extraction (SFE), Standard comparison method (SCM), Spectral library identification (SLI), Retention in-dex (RI), Aroma comparative analysis (ACA), Mass spectrometry (MS), Ion chromatography (IC).

**Table 3 foods-11-02959-t003:** Identification overview for acid substances in different types of baijiu.

Name	CAS Number	Pre-Processing Method	Identification Method	Te-Aroma	Feng-Aroma	Laobaigan-Aroma	Mixed-Aroma	Other Alcoholic Beverages	Tea Flavor	Buckwheat Flavor
formic acid	64-18-6	SPME, LLE	MS, IC, SLI	√	√	—	√	—	—	—
acetic acid	64-19-7	LLE, SDE, SPME, SAFE, DI	MS, RI, ACA, SCM, IC, SLI	√	√	√	√	√	√	√
propanoic acid	1979/9/4	LLE, SDE, DI, SPME	MS, SCM, RI, ACA, IC, SLI	√	√	—	√	√	—	√
2-methylpropanoic acid	79-31-2	LLE, DI, SDE, SPME	MS, SCM, RI, ACA, SLI	√	√	√	—	√	√	—
butanoic acid	107-92-6	LLE, SPME, DI	MS, SCM, RI, ACA, IC, SLI	√	√	√	√	√	√	—
2-methylbutanoic acid	116-53-0	LLE, SPME	MS, SCM, RI, ACA, SLI	—	—	√	√	√	—	—
3-methylbutanoic acid	503-74-2	LLE, SPME	MS, SCM, RI, ACA, SLI	—	—	√	√	√	—	—
tannic acid	1401-55-4	LLE, SDE, SPME, DI	MS, SCM, RI, SLI	√	√	√	√	√	√	√
pentanoic acid	109-52-4	LLE, SPME, DI, SDE	MS, SCM, RI, ACA, IC, SLI	√	√	√	√	√	√	√
2-methylpentanoic acid	97-61-0	LLE	MS, SCM, RI, SLI	—	—	—	—	√	—	—
4-methylpentanoic acid	646-07-1	LLE, DI, SPME	MS, RI, ACA, SLI	—	—	—	—	√	—	—
hexanoic acid	142-62-1	LLE, SPME, DI, SPE, SDE	MS, SCM, RI, ACA, SLI	√	√	√	√	√	√	√
heptanoic acid	111-14-8	LLE, SPME, DI, SPE	MS, SCM, RI, ACA, SLI	√	√	√	√	√	√	√
octanoic acid	124-07-2	LLE, SPME, DI, SDE	MS, SCM, RI, ACA, SLI	—	√	√	—	√	√	√
isooctanoic acid	25103-52-0	SPME	MS, SLI	—	—	—	—	√	—	—
nonanoic acid	112-05-0	LLE, SPME	MS, SCM, RI, ACA, SLI	—	—	√	—	—	√	—
decanoic acid	334-48-5	LLE, SDE, DI, SPME	MS, SCM, RI, ACA, SLI	—	—	√	—	√	√	—
undecanoic acid	112-37-8	LLE	MS, SCM, RI, SLI	—	—	—	—	—	—	—
lauric acid	143-07-7	LLE, SDE, SPME	MS, SCM, RI, MS, SLI	—	—	—	—	√	√	—
tridecanoic acid	638-53-9	LLE	MS, SCM, RI, SLI	—	—	—	—	—	—	—
tetra-decanoic acid	544-63-8	LLE, SPE, SPME	MS, SCM, RI, SLI	—	—	—	—	√	√	—
hexadecanoic acid	1957/10/3	LLE, SPME, SDE, DI	MS, SCM, RI, SLI	—	—	—	—	√	√	—
heptandecanoic acid	506-12-7	LLE, SDE	MS, SLI	—	—	—	—	√	—	—
octadecanoic acid	107-93-7	LLE, SDE, DI, SAFE	MS, SCM, RI, SLI	√	—	—	—	√	—	—
crotonic acid	3724-65-0	SDE	MS, SLI	—	—	—	—	√	—	—
2-methyl-2-pentenoic acid	3142-72-1	SPME, LLE	MS, RI, SLI	—	—	—	—	√	—	—
4-hexenoic acid	35194-36-6	SPME	MS, SLI	—	—	—	—	√	—	—
sorbic acid	110-44-1	SPME	MS, SLI	—	—	—	√	√	—	—
2-dodecanoic acid	143-07-7	SPME	MS, SLI	—	—	—	—	—	—	—
palmitoleic acid	373-49-9	DI, LLE	MS, SCM, SLI	√	—	—	—	—	—	—
linoleic acid	60-33-3	LLE, DI	MS, SCM, SLI	—	—	—	—	—	√	—
oleic acid	112-80-1	LLE, SDE	MS, RI, SCM, SLI	—	—	—	—	√	—	√
vaccenic acid	506-17-2	LLE	MS, SLI	—	—	—	—	—	—	—
(Z)-13-docosenoic acid	112-86-7	LLE	MS, SLI	—	—	—	—	—	—	—
oxalic acid	144-62-7	LLE, SPME	IC	—	—	—	√	√	—	—
pimelic acid	111-16-0	SPME	MS, SLI	—	—	—	—	—	√	—
octane-dioic acid	505-48-6	LLE, SPME	MS, SLI	—	—	—	—	—	√	—
azelaic acid	123-99-9	SPME	MS, SLI	—	—	—	√	—	√	—
(E, E)-2,5-dimethylmuconicacid	20514-41-4	SPME	MS, SLI	—	—	—	—	—	—	—
L-malic acid	97-67-6	DI	MS, SLI	—	—	—	—	√	—	—
D-tartaric acid	526-83-0	SPME	MS, IC, SLI	—	—	—	—	√	—	—
1-propene-1,2,3-tricarboxylic acid	499-12-7	SPME	MS, SLI	—	—	—	—	—	—	—
citric acid monohydrate	5949-29-1	LLE, SPME	IC	—	—	—	√	√	—	—
citric acid	77-92-9	DI, SPME	MS, SLI	—	—	—	—	√	—	—
lactic acid	50-21-5	LLE, DI, SPME	MS, RI, IC, SLI	√	√	—	√	√	—	—
(R)-2-hydroxypropionic acid	10326-41-7	SPME	MS, SLI	—	—	—	—	—	—	—
butanoic acid,2-hydroxy-3-methyl	17407-55-5	SPME	MS, SLI	—	—	—	—	√	—	—
6-hydroxyhexanoic acid	1191-25-9	LLE	MS, SLI	—	—	—	—	—	√	—
7-hydroxyheptanoic acid	3710-42-7	LLE	MS, SLI	—	—	—	—	—	√	—
2-hydroxytetradecanoic acid	2507-55-3	SPME	MS, RI, SLI	—	—	—	—	√	—	—
benzoic acid	65-85-0	LLE, SPME	MS, SCM, RI, IC, SLI	√	—	—	—	√	√	√
3-hydroxy-4-methoxybenzoic acid	645-08-9	DI	MS, SLI	—	—	—	—	√	—	—
vanillic acid	121-34-6	LLE, DI, SPE	MS, SLI	—	—	—	—	—	√	—
salicylic acid	69-72-7	LLE	MS, SLI	—	—	—	—	√	—	—
phenylacetic acid	103-82-2	LLE, SAFE	MS, RI, SCM, SLI	—	—	—	—	—	√	—
4-hydroxy-3-methoxyphenylacetic acid	306-08-1	LLE, DI	MS, SLI	—	—	—	—	√	—	—
4-hydroxycinnamic acid	7400-08-0	LLE	MS, SLI	—	—	—	—	√	—	—
(E)-3-(2-hydroxyphenyl)-2-propenoic acid	614-60-8	DI	MS, SLI	—	—	—	—	√	—	—
3-phenylpropanoic acid	501-52-0	LLE, DI, SPME	MS, SLI, RI, SCM	—	—	—	—	√	—	—
3,4-dihydroxycinnamic acid	331-39-5	SPE	MS, SLI	—	—	—	—	√	—	—
pyruvic acid	127-17-3	SPME	MS, IC, SLI	—	—	—	√	—	—	—

“—” means no relevant data from reference; “√” means have relevant data from reference. The data of Table 3 are collectively derived ([1,5,6,7,8,9,16,17,18,19,20,21,22,23,24,25,26,27,28,29,30,31,32,33,34,35,36,37,38,39,40,41,42,43]). Direct injection (DI), Liquid-liquid extraction (LLE), Liquid-liquid microextraction (LLME), Solid phase extraction (SPE), Solid phase microextraction (SPME), Stir bar sorptive extraction (SBSE), Simultaneous distillation and extraction (SDE), Supercritical fluid ex-traction (SFE), Standard comparison method (SCM), Spectral library identification (SLI), Retention in-dex (RI), Aroma comparative analysis (ACA), Mass spec-trometry (MS), Ion chromatography (IC).

**Table 4 foods-11-02959-t004:** Aroma evaluation for the important acid substances in baijiu.

Aroma Compounds	Cas Number	Aroma Threshold (μg/L)	Aroma Descriptors	Zhima-Aroma						Mixed-Aroma
				Jing Zhi		Fu Tan Chun	Mei Lan Chun	Guo Jing		Lang Ya Tai
				OAV	FD Factor	OAV	OAV	OAV	FD Factor	OAV
acetic acid	64-19-7	160,000	acid, fruit, pungent, sour, vinegar	2	5	11.4	9.69	1	27	5
propionic acid	79-09-4	18,100	fat, fruit, pungent, silage, soy	—	—	—	—	—	1	—
butanoic acid	107-92-6	964	butter, cheese, sour	81	10	410.94	31.75	30	27	350
hexanoic acid	142-62-1	2520	cheese, oil, pungent, sour	3	50		47.83	36	81	38
pentanoic acid	109-52-4	389	cheese, pungent	30		26.5	6.95	27	729	35
heptanoic acid	111-14-8	13,800	apricot, floral, sour	—	5	—	—	—	243	—
nonanoic acid	112-05-0	3560	fat, green, sour	—	—	—	—	—	—	—
octanoic acid	124-07-2	2700	cheese, fat, grass, oil	—	—	—	—	1	27	—
decanoic acid	334-48-5	13,737	dust, fat, grass	—	—	—	—	—	—	<1
hydro-cinnamic acid	501-52-0		fragrant, sweet	—	—	—	—	—	81	—
*α*-methyl-propionic acid	79-31-2	1580	burnt, butter, cheese, sweat	226	—	—	—	3	1	—
4-methylpentanoic acid	646-07-1	144	Floral	—	—	—	—	3	3	<1
2-methylbutanoic acid	116-53-0	5932	butter, cheese, fermented, sour	—	—	—	—	—	—	—
3-methylbutanoic acid	503-74-2	1045	cheese, pungent	45	100	—	—	6	81	—

OAV means odor activity value; FD factor means flavor dilution factor; “—” means no relevant data from reference. The data of Table 4 are collectively derived ([19,21,22,24,27,28,31,32,34,35,36,37,42,43,45,46,47,48,49,50]).

**Table 5 foods-11-02959-t005:** Aroma evaluation for the important acid substances in baijiu.

Aroma Compounds	CAS Number	Aroma Threshold (μg/L)	Aroma Descriptors	Strong-Aroma						
				Zhi Jiang Yuan	Gu Jing Gong		Lu Zhou Lao Jiao	Yang He Da Qu	Wu Liang Ye	Jian Nan Chun
				OAV	FD Factor	OAV	OAV	FD Factor	FD Factor	FD Factor
acetic acid	64-19-7	160,000	acid, fruit, pungent, sour, vinegar	2.78	243	3	5	4	—	8
propionic acid	79-09-4	18,100	fat, fruit, pungent, silage, soy	0.59	—	3	1	—	—	8
butanoic acid	107-92-6	964	butter, cheese, sour	380.22	2187	209	307	64	16	512
hexanoic acid	142-62-1	2520	cheese, oil, pungent, sour	143.84	19683	316	447	64	1024	1024
pentanoic acid	109-52-4	389	cheese, pungent	49.96	2187	89	146	32	256	128
heptanoic acid	111-14-8	13,800	apricot, floral, sour	0.59	27	1	2	64	1024	8
nonanoic acid	112-05-0	3560	fat, green, sour	—	3	<1	—	—	—	—
octanoic acid	124-07-2	2700	cheese, fat, grass, oil	5.44	—	2	10	—	16	8
decanoic acid	334-48-5	13,737	dust, fat, grass	—	—	—	—	—	—	—
hydro-cinnamic acid	501-52-0		fragrant, sweet	—	—	1	1	—	—	—
*α*-methyl-propionic acid	79-31-2	1580	burnt, butter, cheese, sweat	—	—	—	11	8	16	2
4-methylpentanoic acid	646-07-1	144	Floral	—	27	12	9	—	16	8
2-methylbutanoic acid	116-53-0	5932	butter, cheese, fermented, sour	—	—	—	—	—	—	—
3-methylbutanoic acid	503-74-2	1045	cheese, pungent	27.03	6561	18	13	16	—	128

OAV means odor activity value; FD factor means flavor dilution factor; “—” means no relevant data from reference. The data of Table 5 are collectively derived ([19,21,22,24,27,28,31,32,34,35,36,37,42,43,45,46,47,48,49,50]).

**Table 6 foods-11-02959-t006:** Aroma evaluation for the important acid substances in baijiu.

Aroma Compounds	CAS Number	Aroma Threshold (μg/L)	Aroma Descriptors	Jiang-Aroma			Laobaigan-Aroma	Chi-Aroma
				Lai Mao	Mao Tai		Lao Bai Gan	Yu Bing Shao
				FD Factor	OAV	FD Factor	FD Factor	FD Factor
acetic acid	64-19-7	160,000	acid, fruit, pungent, sour, vinegar	256	—	—	—	81
propionic acid	79-09-4	18,100	fat, fruit, pungent, silage, soy	64	—	64	—	243
butanoic acid	107-92-6	964	butter, cheese, sour	256	45	4	27	243
hexanoic acid	142-62-1	2520	cheese, oil, pungent, sour	1616	5	8	27	—
pentanoic acid	109-52-4	389	cheese, pungent	16	62	64	—	81
heptanoic acid	111-14-8	13,800	apricot, floral, sour	—	—	4	9	3
nonanoic acid	112-05-0	3560	fat, green, sour	—	—	4	—	—
octanoic acid	124-07-2	2700	cheese, fat, grass, oil	—	<1	4	—	9
decanoic acid	334-48-5	13,737	dust, fat, grass	—	—	8	—	—
hydro-cinnamic acid	501-52-0		fragrant, sweet	—	—	—	—	—
*α*-methyl-propionic acid	79-31-2	1580	burnt, butter, cheese, sweat	256	—	32	—	9
4-methylpentanoic acid	646-07-1	144	Floral	—	5	8	—	—
2-methylbutanoic acid	116-53-0	5932	butter, cheese, fermented, sour	—	—	—	—	—
3-methylbutanoic acid	503-74-2	1045	cheese, pungent	1024	13	32	9	9

OAV means odor activity value; FD factor means flavor dilution factor; “—” means no relevant data from reference. The data of Table 6 are collectively derived ([19,21,22,24,27,28,31,32,34,35,36,37,42,43,45,46,47,48,49,50]).

## Data Availability

Not applicable.

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
