# Peer review of "“Key Factor” for Baijiu Quality: Research Progress on Acid Substances in Baijiu"

_foods, 2022, doi:10.3390/foods11192959_

Round 1

Reviewer 1 Report

Wu et al. have made a systematic review on importance of acid substances in Baijiu, a national liquor of China by evaluating if acid substances can be decision makers for baijiu quality, their possible contribution to taste and their role in regulating fermentation process in baijiu. This review article is more focused and covered all the key aspects fulfilling aim and scope of this review.

1.      It will be better to include a small overview on recent researches (within last 3 years) in biological activity associated acid substances contained in Baijiu. This sub-title can be included after the “introduction” to lay more emphasis on importance of acid substances in Baijiu as the readers go further into the details of the review. This subtitle can be associated with a schematic diagram to illustrate the possible health benefits of acid substances contained in Baijiu.

2.      All the references in the reference section should conform to the “Foods” journal referencing style.

3.      All the abbreviations used in both figures and tables should be described in full form under the respective figure caption or table footnotes.

4.      Conclusion should be of one single paragraph.

Reviewer 2 Report

I reviewed the manuscript entitled, “Decision Maker” for Baijiu Quality: Research Progress on Acid Substances in Baijiu. The manuscript focused on acid substances in Baijiu by highlighting the research conducted on Baijiu. The manuscript is well written and cited the literature up-to-date. The comments/suggestions:

Decision Maker is not appropriate in title

Line 13: passed down; kindly use more relevant and appropriate word

Abstract should be revised highlighting review objectives, review findings, conclusions, and future recommendations

Table 1. Any disadvantages of each pre-treatment method should be provided

Line 114 and 115: provide the list of analytical methods used in Table form

Figure 2 quality must be improved

Figure 2c can be converted to Table form (for functional applications)

References are not according to the journal format. Please revise it.
